# Hard and Soft Protein Corona of Nanomaterials: Analysis and Relevance

**DOI:** 10.3390/nano11040888

**Published:** 2021-03-31

**Authors:** Rafaela García-Álvarez, María Vallet-Regí

**Affiliations:** 1Departamento Química en Ciencias Farmaceúticas, Unidad de Química Inorgánica y Bioinorgánica, Instituto de Investigación Sanitaria Hospital 12 de Octubre i+12, Universidad Complutense de Madrid, Plaza Ramón y Cajal s/n, 28040 Madrid, Spain; 2CIBER de Bioingeniería, Biomateriales y Nanomedicina, CIBER-BBN, 28029 Madrid, Spain

**Keywords:** protein corona, hard corona, soft corona, analytical techniques

## Abstract

Upon contact with a biological milieu, nanomaterials tend to interact with biomolecules present in the media, especially proteins, leading to the formation of the so-called “protein corona”. As a result of these nanomaterial–protein interactions, the bio-identity of the nanomaterial is altered, which is translated into modifications of its behavior, fate, and pharmacological profile. For biomedical applications, it is fundamental to understand the biological behavior of nanomaterials prior to any clinical translation. For these reasons, during the last decade, numerous publications have been focused on the investigation of the protein corona of many different types of nanomaterials. Interestingly, it has been demonstrated that the structure of the protein corona can be divided into hard and soft corona, depending on the affinity of the proteins for the nanoparticle surface. In the present document, we explore the differences between these two protein coronas, review the analysis techniques used for their assessment, and reflect on their relevance for medical purposes.

## 1. Introduction

Nanomaterials (NMs) designed for biomedical applications such as drug delivery [1,2,3,4,5], or for therapy/diagnosis [6,7,8,9] are expected to come into contact with physiological media, and therefore with a wide variety of biological entities present within the body [10]. Biomolecules such as lipids, proteins, or nucleic acid fragments, as well as exosomes or cells are very likely to interact with the injected NM, leading to the modification of its physico-chemical characteristics, and thus its behavior and fate after administration [11]. Amid the extensive range of bio-interactions, protein adsorption is considered as one of the most relevant due to the high amount of this biomolecule type in physiological fluids [12]. Consequently, it has been the object of intense study for a long time.

The protein adsorption phenomenon was described for the first time in 1962, in a study of interactions between hydrophobic powders and plasma samples performed by L. Vroman [13]. In this publication, Vroman reported a change in the surface properties of the starting material, which becomes hydrophilic after interaction with plasma. It is important to notice that the modifications on the wettability of the material are not exclusively caused by protein interactions, but also by the initial physico-chemical characteristics of the powders. These interesting results lead to decades of research in interactions between proteins and different types of materials and NMs [14].

It was not until 2007 that Cedervall and co-workers [15] introduced the term “protein corona” (PC) to describe the resulting dynamic protein structure formed on the surface of NMs upon incubation in a biological milieu (Figure 1). In their publication, Cedervall and colleagues presented the first systematic investigation of PC on NMs, stating its importance for the bio-identity of the NM, and hence, its potential for biomedical applications. The authors proposed isothermal titration calorimetry as a suitable technique for the study of affinity and stoichiometry of proteins towards nanoparticles (NPs), and size exclusion chromatography (SEC) as a less perturbing method of PC–NP complex separation.

Countless efforts have been invested in deeper study of the PC formation mechanisms [16,17], its characterization [18,19,20], the effect of NM characteristics on the PC [21,22,23], as well as the influence of the PC itself on a variety of biological processes [24,25,26,27]. Nowadays, PC is considered as a key parameter that must be investigated and understood for every NM proposed for biomedical applications before any translation into clinical trials [28,29,30].

In the present document, we review the current knowledge about the dynamics and kinetics of protein adsorption that lead to PC formation, the existence and relevance of the hard and soft PCs, as well as the different strategies and techniques that are used for their isolation and study.

## 2. Protein–Nanomaterial Interactions

Protein–NM interaction is a highly complex phenomenon determined by a wide variety of parameters such as the dynamicity of the environment, the physico-chemical characteristics of the NM, and the structure of proteins, among many others [31,32]. Protein adsorption is the result of the interactions between the NM surface, proteins and other biomolecules present in the media. These interactions are mainly driven by non-covalent forces, affinity constants of proteins, and protein structure thermodynamics [33,34]. In the next sections, we will explore the forces involved at the protein–NP interface as well as the kinetics of adsorption guiding these interactions.

### 2.1. Forces at the Interface

Protein adsorption on a solid surface (e.g., an NM) is a very common but complex phenomenon that involves a wide variety of different variables [35]. Although incubation conditions and NM surface play an essential role on NM–protein interactions, the majority of interactions that drive protein adsorption are directly related with maintaining the tridimensional conformation of proteins themselves [36]. Due to their nature, proteins are susceptible to a wide range of interactions (Figure 2): (i) they can form strong and stable covalent bonds with other molecules; (ii) they can establish electrostatic interactions due to the presence of charges on their structure; (iii) due to the presence of H atoms linked to very electronegative atoms, they can readily create hydrogen bonds with other materials or molecules; (iv) they can display weaker but equally relevant interactions such as Van der Waals forces, hydrophilic/hydrophobic interactions, and contact with other ions of the solvent [37].

In the case of protein adsorption on NMs, electrostatic interactions are known to play an important role [38]. It has been demonstrated that the surface of the NM, the pH of the media, and the charge of the protein are fundamental for the formation of the PC. In fact, it is well known that neutral NMs tend to interact less with proteins in comparison with their charged counterparts [21]. Several studies use electrostatic interactions and the formation of electric layers as an explanation for the interaction of proteins of both positive and negative charges, with charged NM surfaces considering a sequential binding [39]. For example, in the case of a positively charged NM, one would expect interaction with negatively charged proteins more than with any others. However, after an initial layer of proteins is adsorbed on the NM surface, the superficial charge of the NM is modified to more neutral or differently charged, giving the opportunity of interactions with neutral, positive, and negatively charged proteins.

Amid the parameters that can influence processes dominated by electrostatic interaction, pH is considered a very important one [40]. Protein tridimensional structure and its global charge are dictated by the pH of the media. It is well known that proteins generally tend to be pH-dependent in terms of protein adsorption. This is due to fact that the isoelectric point of proteins is modified as a function of the pH. Although this would imply that the protein interactions could be controlled by the pH, this is not completely true due to the complexity of real plasma and the high diversity of proteins present.

It is true that electrostatics are essential for protein unfolding, and therefore, in protein adsorption. However, it must be kept in mind that electrostatic attraction can be easily minimized by the optimized action of a number of weaker interactions [33,41]. For the adequate understanding of protein adsorption, it is fundamental to take into account the full set of forces at the nano-biointerface.

### 2.2. Protein Adsorption on Nanomaterials

The dynamic process of adsorption–desorption can be described in mathematical terms, which help to predict the interactions of NM in the presence of proteins [42]. Parameters such as the kinetic constant of adsorption/desorption processes as well as the diffusion constant of proteins can be used to infer the reversibility degree of the interaction and the affinity of the proteins for the NM surface. The kinetics of protein adsorption on a surface can be expressed by the following equation:$$P+S\;\underset{k_{des\;}}{\leftarrow }\overset{k_{ads}}{\to }P_{s}$$
where *P* represents the proteins, *S* represents the surface of the NM, and *P_S_* represents the complex formed after their interaction. The terms *k_ads_* and *k_des_* denote the kinetic constants of adsorption and desorption, respectively. It is important to highlight that this process is reversible, and that the degree of reversibility depends on the values of the kinetic constants *k_ads_* and *k_des_*. The reaction would become irreversible when *k_des_* = 0, which limits the kinetics of this equation.

In order to describe the protein–NM interaction, we must take a closer look at the different steps of the process itself. In the first place, when an NM is incubated in a biological fluid, both entities are subjected to motion forces such as diffusion or convection, which transports them and help them come into contact with each other. In addition, parameters such as the diffusion constant, the interfacial concentration, or the protein concentration are also to be kept in mind. Next, proteins would interact with the NM surface by different forces and binding types. Finally, these interactions will lead to the attachment of the protein on a specific part of the NM.

Typically, the Langmuir model (Figure 3) has been widely used for the study of protein adsorption [43]. In this model [44], it is assumed that the NM surfaces have a limited number of adsorption sites, all being energetically equivalent. Moreover, the process is considered reversible where, at the equilibrium state, adsorption and desorption phenomena occur and binding sites are occupied, forming a monolayer. Nevertheless, this model is not enough to describe PC formation [43], because it is well known that some proteins are irreversibly attached to the NM surface, protein adsorbed can undergo unfolding and reorientation, and the NM surface is rather heterogeneous in most cases. Additionally, PC is known to be formed by several layers of proteins, being a multi-layered structure of high dynamicity, where not only protein–NM interactions are important, but also protein–protein interactions play an essential role [30].

Numerous scientists have proposed different models for the description of this process, taking into account parameters such as the heterogeneity of the surface [45,46], the saturation point [47], or the protein structure and reorientation after undergoing adsorption [48]. However, the crucial problem of the reversibility of the process was rarely taken into account in any of these models. Due to the complexity of the field, researchers are still working to obtain a deeper understanding and to design more adequate models for the investigation of this process. In the meantime, the Langmuir model is still the most widely employed model for the description and study of the protein adsorption process.

### 2.3. Avoiding the Protein Corona

Surface functionalization of NMs is the most common strategy used to control their interactions with proteins [49]. Although one could take advantage of the PC for applications such as biomarker discovery [50,51] or increased drug loading [52,53], the process of PC formation is commonly considered quite problematic and scientists have invested efforts on designing and engineering coatings that can avoid, as much as possible, protein adsorption on the NM surface. The objective of this strategy is not only to confer a “stealth” character to the NM, but also helps to elude its opsonization by immune cells once within the bloodstream, extending its circulation lifetime in this way.

Among the great variety of available molecules that could be used for this purpose, zwitterionic structures [54,55,56] as well as different types of polymers have been investigated [49,57]. These molecules have demonstrated to be very useful for controlling protein adsorption onto NMs. These coatings are able to minimize the interactions between proteins and NMs, which lead to a decrease in the amount of protein adsorbed on the NM surface (Figure 4).

In this regard, polyethyleneglycol (PEG) has been widely used for biomedical applications thanks to its biocompatibility [58,59] and antifouling capability [60,61]. Extensively employed for in vivo experiments, it has been reported to cause relatively low toxicity while displaying high solubility and stability in water. Moreover, it avoids opsonization, increasing the time the NM has for interacting within the body. Several studies have reported that the PEGylation of NPs reduces protein adsorption upon incubation in physiological fluids such as blood, plasma, or protein mixture solutions [60,61]. In addition, it has been demonstrated that the molecular weight of the polymer plays an important role on corona formation, the latter being lower when the molecular weight is increased. However, PEG cannot fully avoid the formation of a PC around the NM. It is interesting to note that some researchers have suggested the need of certain protein adsorption for this “stealth” effect [62]. In their publication, Schötler and his team synthesized polystyrene NPs and further functionalized them with PEG and another polymer called polyethylethylenephosphate. These NPs were exposed to plasma proteins and their PCs were investigated. Results showed that the presence of these polymers could be used for minimizing protein–NP interactions, and therefore reduce the PC formed around the NPs. Remarkably, authors suggested that the “stealth” property of the NM seemed to be related to the abundance of a specific protein in the corona, clusterin [63].

On the other hand, zwitterionic structures have also attracted a lot of attention because they display great antifouling properties that can be applied not only to minimize protein adsorption on NMs [54,55,64], but also to avoid bacterial adhesion [65,66,67] or biofilm formation [68,69,70] on biomedical devices such as implants. These polymers are characterized by having equal anion and cation groups on their chains, which gives them very high hydrophilicity and interesting antifouling properties. Although of remarkable performance, the utilization of PEG is still more generalized.

## 3. The Structure of the Protein Corona

PC is currently described as a dynamic multi-layered structure formed by proteins adsorbed onto a NM surface upon contact with the physiological environment and consequent interaction with proteins [12]. This structure can be generally divided in two parts, known as “hard” and “soft” PCs. While the inner layer of tightly bound proteins with a longer lifetime has been termed as “hard” corona (HC), the outer layer of weakly bound proteins with a shorter lifetime is called “soft” corona (SC) [71]. Both of them contribute to the new bio-identity of the NM after protein adsorption; therefore, it is fundamental to understand them in the case of NMs designed for biomedical applications. The dynamic nature of the PC as well as information and characteristics of these structures will be discussed next.

### 3.1. Hard vs. Soft Protein Corona

As mentioned before, protein adsorption to an NM surface is a complex phenomenon in terms of the numerous parameters that need to be considered for an adequate description of the process [35]. Its high dynamic nature is due to the numerous physico-chemical interactions as well as the thermodynamic exchanges between NM and proteins and, also, between proteins themselves [33]. At the moment of injection or incubation of the NM in the biological fluid, the conditions of the experiment along with the NM surface and protein kinetics play an important role, defining the formation of the PC.

The dynamic nature of the phenomenon can be best described by using the “hard” and “soft” corona terms [71], and their main differences can be observed in Figure 5. The HC is known to be constituted by proteins with high affinity for the NM surface, which translates into a rapid formation of a tight, strong, NP–protein bonding that leads to the formation of a more stable complex over time. These properties are very helpful when analyzing this particular structure, because it is consequently possible to isolate and characterize by using both in situ and ex situ experimental procedures [18]. For these reasons, there are a great number of publications which describe the HC of a wide variety of NMs such as liposomes [72,73,74], quantum dots [75], metallic NPs [39,76,77], silica NPs [78,79], polymeric NPs [15,27,80], and 2D materials [81], and many more.

The existence of the HC has been a widely discussed topic, especially when this structure was first proposed; hence, it has been extensively investigated. Several publications have reported the formation of an irreversible protein adsorption on the NM surface upon incubation in a physiological fluid [74,82,83]. Although initial studies were performed using single protein solutions, later investigations were carried out using more complex media such as plasma or blood. Results indicate, in general, that in the majority of cases, proteins interact with the NM surface in an irreversible way. Nevertheless, it is important to keep in mind that in certain cases, at a particular NM size and morphology, using the adequate coating and in some single protein solutions, minimal protein adsorption takes place, leading to an incomplete covering of the surface of the NM [84].

The time-evolution of the PC has been studied in order to demonstrate the existence of the HC. It was in 2010 when the evolution of a loosely formed corona towards an irreversible one was observed [83]. In this study, Casals et al. observed the PC form on the surface of citrate-capped gold NPs after incubation in cell media. Additionally, Pisani and coworkers [78] observed a similar behavior for the protein adsorption process on the surface of silica NPs exposed to a biological fluid. In this work, the authors combined mass spectrometry with computational biology in order to understand the growth of the PC and to define what they called “interactome”, which would refer to interactions among proteins of the corona. Remarkably, a study developed by Hadjidemetriou et al. [74] enabled the investigation of the in vivo PC for the first time (Figure 6). In their study, lipid nano-vesicles were injected in mice and later recovered at different time points by cardiac puncture. A combination of SEC and membrane ultrafiltration was employed for the separation and purification of PC–NP complexes. Analysis by mass spectrometry (MS) revealed that a complex corona was already formed as early as 10 min post-injection. The authors found that the composition of the PC varied over time. Finally, Weiss et al. [17] have reported the in situ characterization of the time-evolution of the PC formed around silica microparticles under flow conditions. Their results revealed that PC is kinetically divided into three phases: (i) proteins are adsorbed irreversibly to the NM surface; (ii) these irreversibly bound proteins interact with pre-adsorbed ones; and (iii) reversible binding of SC proteins. A combination of confocal laser scanning microscopy and microfluidics was employed for the analysis.

On the other hand, the SC has been one of the most complex topics of study of the PC field. This corona is said to be constituted by proteins of a lower affinity for the NM surface, and therefore it takes more time for its constitution and is more unstable [71]. In addition, due to the dynamic nature of the PC, most outer proteins are believed to be constantly exchanged as response to the conditions of the media. Of unstable nature and difficult to isolate, its behavior is usually observed by using in situ methodologies and analytical techniques, because it cannot be isolated in the same way as the HC [19,20]. It is important to highlight that protein–protein interactions play a much more important role in the formation of the SC, because the NM surface has been already occupied by other proteins. Additionally, the electrostatic response will be dependent on the new surface charge of the protein-coated NM and affected by steric interactions of the proteins adsorbed. These particularly complex characteristics make the SC of NMs even more difficult to investigate.

Nevertheless, scientists are using different approaches to try to isolate the SC. For example, Weber [85] and coworkers employed a combination of centrifugation and asymmetric flow field-fractionation for the separation and further study of the SC formed on the surface of polystyrene NPs. Interestingly, this study indicates that, in their particular system, not the SC but only the HC had an effect on cell uptake of these NPs. In a different study [86], however, the authors performed an in situ analysis of the corona on liposomes upon incubation in plasma. In this case, results suggested that SC contributed to the stealth properties of the NM.

### 3.2. The Exception—Nanomaterial Size = Protein Size

Up to now, studies have mainly been focused on results from investigations on NMs of larger dimensions than proteins in the media (≥40–200 nm size). However, some studies suggest that PC formation is different when the NM and protein are of a similar size. Interestingly, there are few investigations dedicated to this particular case. The first indications of these remarkable differences were observed by Casals et al. [83], where 4 nm gold NPs did not present a formation of what is known as an HC, but instead were revealed to weakly interact with proteins in the media.

With the purpose of developing a deeper understanding of these differences, Liu and colleagues [84] evaluated protein adsorption on cerium oxide NPs of 7 and 9 nm diameter (Figure 7). Contrary to the considerations taken for large NPs, the surface of the NM with respect to the protein cannot be regarded as flat and infinite any more for very small NPs. Results revealed that these small NPs interact with proteins in a reversible way, which means that no HC is formed, only SC. In addition, the authors indicate that the interaction between NPs and cells may be modified, but different PCs formed on NPs are of different sizes. In the case of larger NMs, protein adsorption of certain proteins is irreversible, leading to HC formation and changing the bio-identity of the NM. Nevertheless, this would be different for very small NMs, where there are more chances of the cell membrane interacting with the “bare” NP, because the process of protein adsorption is reversible. This information is relevant and must be taken into account for future investigations, where a different model of protein adsorption on NM surfaces must be applied.

## 4. Parameters Affecting Protein Corona

Due to the complexity of interactions at the bio–nano interface, PC formation is known to be affected by a wide variety of parameters as well as by having influence on many different biological processes [28]. Physico-chemical characteristics of NMs such as size, shape, surface charge, or functionalization are important factors that must be taken into consideration when investigating PC formation. Similarly, choice of incubation media is also essential, because it will directly impact on PC composition. Equally relevant are the experimental conditions such as temperature, exposure time, or dynamic/static flow conditions.

### 4.1. Nanoparticle Properties and Protein Corona

Regarding the influence on PC formation, size has been one of the most widely investigated parameters. Dobrovolskaia et al. [39] dedicated their efforts to the study of the PC formed on the surface of citrate-capped AuNPs incubated in human plasma. In this investigation, they examined the PC formed on AuNPs of 30 and 50 nm, reporting that a higher amount of protein was adsorbed on the smaller NP. In a similar study, Tenzer and co-workers [87] incubated SiNPs of different sizes in blood plasma samples. Their results were in agreement with Dobrovolskaia study, showing an increase in protein adsorption for smaller NPs. Size influence on PC has also been investigated for co-polymeric NPs. In their publication, Lundqvist et al. [21] performed an investigation on polystyrene NPs of 50 and 100 nm diameter upon exposure to human plasma. In agreement with previous investigations, their results indicated that NP size plays an important role in protein adsorption, modifying binding constants of proteins and leading to a different PC composition. In this last publication, although protein profiles of NPs were quite similar for all coronas, some specific proteins were present only in the larger or smaller counterpart.

Shape is an important factor that has somehow been overlooked in PC studies, which is reflected in the low number of publications related to PC dependence on NP morphology with a detailed characterization of its composition. Deng et al. [77] investigated PC formation in titanium oxide NPs of various morphologies: nanospheres, nanorods and nanotubes. The authors found that certain proteins were present in PCs of spherical NPs, which suggests an influence on shape on protein binding affinity. In an analogous study, Ma and co-workers [22] investigated the PCs of spherical and rod-like SiNPs after incubation in single protein solutions. However, the authors do not report the potential biological impact of the presence or absence of certain proteins on the different PCs. Their results demonstrated that some morphologies could cause conformational modifications on some proteins. Lastly, the PCs of AuNPs formed in vivo of different size and shape were analyzed after injection in mice [88]. Results indicated that both parameters, size and shape, had an effect on PCs that led to different amounts of protein adsorbed and PCs of different composition. This study suggested that the total amount of protein adsorbed does not necessarily reflect the complexity of PC composition.

Another crucial factor that could impact NM–protein interactions is the surface charge of the NM. Positively charged NMs are known to be rapidly recognized by opsonins, which lead to their elimination from the body and later accumulation in the liver and spleen [89]. In order to prevent opsonization, NMs are usually functionalized with negatively charged molecules or polymers to display a zeta potential value of around −50 to −30 mV [90]. PC studies on positive, neutral, and negatively charged AuNPs were performed by Lundqvist et al. [21], leading to the conclusion that surface charge is a fundamental factor for protein adsorption. Authors discovered that neutral NPs seem to adsorb a smaller amount of protein and of a lower variety in comparison with their charged counterparts. Nevertheless, authors did not examine or predict the possible biological impact of the absence or presence of certain proteins on different PCs. A similar study was performed by Kah et al. [91] on AuNRs of different surface charge. However, they did not report the protein profile of the PCs of the NPs and exclusively focused on the effect of PC-coated NPs on cell internalization and proliferation.

Functionalization is also known to play an essential role on PC formation, because it defines the NM surface, and hence, largely determines protein–NP interactions. This strategy is mainly aimed to confer a stealth character to the NM, which also helps to elude opsonization by immune cells once within the bloodstream. As mentioned before, PEG is one of the most widely used polymers for this purpose and it has been shown to minimize protein adsorption on NMs. Interestingly, the authors report that this stealth property seems to be related to the abundance of a specific protein of the corona, clusterin [62]. In addition, the authors proposed a protein pre-coating of the NP in order to modify its physiological action. Mirsafiee et al. [92] have suggested that a pre-coating of gamma-globulin on SiNPs would increase the amount of opsonin in their PCs, increasing NP uptake by macrophages. Nevertheless, this modification did not lead to an improvement of NP uptake by cells.

Although the NM surface is usually not directly exposed to the biofluid—NM surfaces are generally modified and engineered to exhibit certain physico-chemical properties by functionalization with different molecules—there seems to be an effect of the natural composition of the NM and the PC composition. Even though some comparative studies of PCs of different NPs have been performed, the main differences on their protein profiles have not extensively been explained. However, one could observe some important differences when looking at the literature, in particular, tables of the most abundant proteins reported for different NPs. Lipid-based NPs seem to have a higher tendency to interact with apolipoproteins, as can be seen in several publications by Hadjidemetriou et al. [93] On the other hand, AuNPs exhibit serum albumin, alpha-2-macroglobulin, and fibrinogen chains as the most abundant proteins, which some authors have suggested may be linked to the fibrillation process that some proteins undergo after contact with AuNPs [39,88]. In addition, a study by H. Wang et al. [75] reported alpha-2-macroglobulin and some complement proteins as the most abundant in the PC of quantum dots. Some of the mentioned proteins actively influence colloidal stability and biological interactions of NMs. For example, adsorption of opsonins such as immunoglobulins or complement proteins are known to enhance phagocytosis and consequent removal from the bloodstream. On the contrary, serum albumin and apolipoproteins are known to extend circulation of NMs in the blood. Nevertheless, the prediction of protein–NM interactions, as well as the potential biological impact of NMs by PC analysis, remains a very complicated and difficult task.

### 4.2. Experimental Conditions and Protein Corona

In addition to NP properties, the choice of biofluid is a critical factor that plays a major role in PC composition. The impact of the use of different physiological fluids has been investigated by several groups. One of the first studies focused on this issue was performed by Maiorano and co-workers [94]. They carried out an investigation of PCs formed upon the incubation on two different cell culture media. Their results indicated that the abundance of proteins within the corona did not correspond to the amount of protein on the incubation media, which was explained by taking into account the different binding constants of the proteins present in the media. Interestingly, while incubation in Dulbecco’s modified Eagle’s medium evolved towards a stable PC, the corona formed upon interaction with Roswell Park Memorial Institute media was more reduced and unstable. Although this study was quite complete, the authors did not take into consideration the biomolecules secreted by the cells in a process known as “cell conditioning”. Albanese et al. [95] decided to investigate the influence of this phenomenon on PC formation. Results indicated that cell conditioning produced aggregation as well as some changes on PC composition, related with NP size, surface charge, and cell phenotype.

Apart from cell culture media, serum and plasma have been widely used for PC investigation experiments. The main difference between these two biofluids is that, while plasma is the liquid cell-free part of the blood treated with anticoagulants, serum is the liquid part of blood after coagulation, and thus does not contain clotting factors. An interesting comparative study between the PCs formed on human serum and human plasma was performed by Mirshafiee et al. [96]. Data obtained showed a higher amount of protein adsorbed after incubation in human plasma in comparison with human serum. As expected, some differences in the PC composition were reported, where a higher fraction of complement proteins and coagulant factors were observed for NPs incubated in human plasma. Solorio-Rodríguez and co-workers [97] carried out an important investigation for the comparison of PCs formed upon interaction with mouse plasma and human plasma. Once again, significant differences on PC composition of the coronas were observed. It is important to note that this study is particularly important because it states the limitations of clinical trials performed in animal models. Finally, some authors have reported significant differences on the PCs formed on plasma sample of patients suffering from different diseases, highlighting the potential of PC characterization as a diagnostic tool [98,99,100].

Time evolution of the PC was investigated for the first time in 2010 by Casals et al. [83]. The authors incubated AuNPs of different sizes in cell culture media, monitoring the PC formation over time. Remarkably, results suggested the evolution of the corona from a transient complex into an irreversible structure. Similarly, Pisani et al. [78] investigated the PC formed onto the surface of SiNPs over time to understand the growth of the corona and to define what they called the “interactome”, which would refer to the interactions among the proteins of the corona. Finally, a study carried out by Hadjidemetriou and co-workers [74] allowed the investigation of the time evolution of the in vivo PC for the first time. Lipid NPs were injected in mice and later recovered, and the PC–NP complexes were purified and analyzed. Results revealed the formation of a complex PC as soon as 10 min post-injection. Even though the amount of protein adsorbed was not significantly affected, PC composition was found to vary over time, which indicated the high dynamic character of the corona.

Temperature is another important factor that must be taken into account when studying the PC. It is well known that body temperature varies depending on the part of the body and its activity. Moreover, intracellular temperature is also subject to changes. Consequently, it is necessary to better understand the relationship between PC formation and temperature. Surprisingly, there are very few publications regarding this matter. One of these investigations was performed by Mahmoudi et al. [101], where the authors incubated inorganic NPs on fetal bovine serum (FBS) at a wide range of temperatures (5–45 °C). Results revealed that, although changes in the temperature may significantly modify PC formation and composition, this is not always the case. On many occasions, NMs need to be heated to perform a specific task within the body, for example, plasmonic NPs. In order to determine if plasmonic heating by laser activation affects PC, Mahmoudi and co-workers [102] carried out an investigation by the incubation of AuNRs in FBS solutions and heating by plasmonic and traditional thermal heating. Results indicated significant changes on the PC composition formed around low aspect ratio AuNRs.

Lastly, static or dynamic conditions of the experiment are expected to influence PC formation. In an investigation carried out by Palchetti et al. [73], PC formed on the surface of liposomes after injection in circulating FBS was studied as a reference of the PC formed under in vitro conditions. The main objective of this investigation was to assess the effect of the shear stress produced by the flow on PC formation. Results indicated that corona composition was significantly influenced by the dynamic flow; therefore, the PC formed under in vitro conditions would considerably differ from its in vivo counterpart. Sakulkhu and co-workers were the first to present a full characterization of the PC composition formed on the surface of magnetic NPs under realistic conditions. In this study, NPs were injected into the bloodstream of rats and later recovered by using strong magnets. Results showed significant differences in PC composition, meaning that dynamic conditions strongly influence protein–NP interactions. Similarly, Hadjidemetriou et al. [93] reported a study on the in vivo PCs formed on lipid NPs after injection in mice. Comparative of in vitro and in vivo PC composition was carried out by MS, revealing a wider variety of adsorbed proteins in the corona formed under in vivo conditions. The authors reported that in vitro conditions far from adequately mimic the high complexity of the real physiological environment, and hence they cannot be used for prediction of the in vivo PC.

## 5. Analysis of the Protein Corona

The most extended protocol for the investigation of the formation, physico-chemical characteristics, as well as the composition of the PC is explained in detail by Monopoli and co-workers [18]. In general, the procedure begins with the incubation in a physiological fluid under defined conditions. Next, the NM–PC complex is isolated and purified from free proteins in solution by different techniques, depending on the nature of the NM. Physico-chemical properties of NMs before and after protein interaction are compared, and a proteomic analysis is employed to investigate the PC composition. This strategy could be defined as an ex situ approach of study. Nevertheless, these type of protocols are characterized by focusing on the study of the HC, which is more stable and can be isolated, without giving information about the SC [20].

Due to the relevance of understanding the role of SC on NM–protein and protein–protein interactions and its consequences on other biological processes, researchers have proposed an alternative strategy that allows them to investigate the SC: in situ methodology [20]. Employing this different type of procedure, scientists have been able to observe PC formation and behavior in real time and in the presence of the media, without purification [103,104]. However, this approach does not generally give information about PC composition and needs to meet several requirements such as the presence of a fluorescent tag [80,105] or the use of spherical NPs [20,83].

In the next sections, we will review and discuss the main differences between ex situ and in situ analysis (Figure 8), advantages and disadvantages of both approaches, along with the techniques used during the different steps of the process. Both strategies are very important for an adequate investigation of the PC.

### 5.1. Ex Situ Analysis of the Protein Corona

This kind of experiments is based on an analysis of the PC performed after isolation and purification of the NP–PC complex [18]. Although it is the most commonly utilized procedure for PC investigation and has helped in the understanding of this phenomenon, it fails to give information of the protein adsorption process in real time or details about the SC. During the following sections, we will explore the different steps generally employed for this type of PC study, the techniques utilized for its analysis, as well as the advantages and disadvantages of this strategy.

#### 5.1.1. Incubation Conditions

The first step for the ex situ analysis of the corona is the incubation of the NM on the selected biological fluid. Depending on the final applications of the designed NM, two main incubation approaches can be applied, in vitro and in vivo. Both give useful information about the process under study and have been widely employed. Nevertheless, their different conditions are expected to have an influence on the process of PC formation, leading to distinct results, which must be carefully interpreted.

In vitro incubations are the most commonly performed, because there is no need of utilizing animals for the experiments, which simplifies the methodology and work. Biological fluids used for these strategies such as blood, plasma, protein solutions, simulated body fluid and many more, are readily available for purchase from different companies. In addition, experimental conditions such as temperature, time, or shaking speed, among many others, can be easily controlled and modified. However, the main limitation lies in the fact that these experiments are not an accurate representation of the real conditions, because many parameters such as the flow dynamics and presence of other bio-entities are overlooked [93]. It is important to mention that some researchers have introduced the parameter of the flow rate by using microfluidic chips for flow dynamic experiments [106].

On the other side, in vivo incubations are utilized for more realistic information regarding the PC formation process [93,107]. There are plenty of different options, although mice and rats are the most widely employed models [76]. These animals are available in different strands, which may be more or less adequate depending on the type of experiment to be performed. It is important to note that these experiments must be thoroughly planned and ethically justified. There are several parameters that must also be decided, such as the administration route, amount of sample injected, recovery of the sample, and length of the experiment. The main drawbacks of this strategy are the need of animals, the lack of control over the experiment, and the sample loss.

#### 5.1.2. Nanoparticle–Complex Separation

Once the NMs have been allowed to interact with the biological entities present in a determined physiological media, NM–PC complexes must be separated from unbound free proteins. Several methodologies can be selected, taking into consideration the physico-chemical characteristics of the NM under study. We will explore some of the most common options: traditional centrifugation [18], differential centrifugation sedimentation (DCS) [108], size exclusion chromatography (SEC) [93], and magnetic separation [76].

Centrifugation is the most extensively used technique for the isolation of NM–PC complexes because it can be applied to a wide variety of NMs. Its principle is displayed in Figure 9A. In a publication presented by Monopoli et al. [18], it is described as a tool for corona-coated NMs separation, based on the use of several centrifugation steps for an adequate separation from free-unbound proteins. Considering the convenience, simplicity, and versatility of this approach, it is not surprising that it remains as the most commonly used for PC studies. Nevertheless, this methodology presents some drawbacks, such as the very strong centrifugal forces applied for the separation—especially for very small or low-density NMs—which may influence interactions between the NM surface and proteins, and hence, the stability of the complex. Certainly, this approach fails to isolate the SC [19], because protein–protein interactions present in this structure are too weak to survive the centrifugation step. In addition, it is possible that, depending on the strength of the centrifugation, some free proteins could be pulled down and not really separated from the complexes.

An alternative to typical centrifugation that has been proposed and employed in several PC studies is DCS [18,108,109]. Additionally, known as the two-layer sedimentation method, it has been widely applied for the determination of NM size with extremely high resolution. This technique is based on the different sedimentation times of NPs as a function of their size. A comparison of the performance of traditional centrifugation vs. DCS was carried out by N. Fernández Iglesias et al. [110]. In this investigation, the authors evaluated the PC formed on the surface of spherical gold NPs by normal centrifugation and DCS, and consequently compared the performance of both techniques. Although no significant differences were observed in relation to the qualitative composition of the corona, results suggested a higher efficiency for PC–NPs isolations of DCS over normal centrifugation. Unlike the latter, authors have been able to apply DCS for the isolation of the more elusive and unstable SC. For instance, Davidson et al. [108] investigated the corona of gold NPs formed upon incubation in a solution of bovine serum albumin. Authors employed DCS to detect the HC and SC proteins and described their behavior by using Langmuir-type isotherms.

In some cases, scientists can take advantage of certain properties of determined NMs, as is the case of magnetic NPs. One example is the work of Sakulkhu and co-workers [76], where they investigated the PC formed on the surface of magnetic NPs and employed a strong magnet for the isolation of the PC–NP complexes. Similar to the centrifugation procedure, this initial separation step was followed by several washing steps until no protein was detected in the supernatant.

Although sometimes particular NM properties can be an advantage, in other occasions they can become a drawback. That is the case of lipidic NPs, whose characteristic low density prevents the use of traditional centrifugation for PC–complex purification. Hadjidemetriou and colleagues [93] developed an alternative methodology that allowed the separation of corona complexes based on SEC followed by membrane ultrafiltration. SEC, also known as molecular sieve chromatography, is an analytical technique that leads to the separation of molecules from solution by their size (Figure 9B). Even though this technique has mainly been employed for separating mixtures of molecules of distinct size, it has been successfully applied to the isolation of PC–NP complexes. Nevertheless, Hadjidemetriou et al. reported that this technique must be complemented, in this case, by membrane ultracentrifugation to efficiently separate the complexes from unbound free proteins. Although centrifugation is still needed, the use of these columns for membrane ultrafiltration allows the use of lower centrifugation speeds. Initially developed for lipid NPs, this approach can be optimized for other NMs, such as gold NPs [88].

#### 5.1.3. Protein Corona Analysis

Apart from analysis of modification of the physico-chemical properties of NPs before and after incubation with proteins, it is mandatory to perform a qualitative and quantitative study of the composition of the corona formed on the NM surface [18]. There are several techniques employed for this purpose. In this small review, we will briefly present bicinchoninic acid assay (BCA assay), gel electrophoresis, and nano liquid chromatography tandem mass spectrometry (LC-MS/MS) and its use for PC analysis (Figure 10).

BCA assay is a very popular method employed for the determination of the total protein amount present on a sample, and it has been employed for the quantification of the amount of protein of the corona [111]. The principle of this approach is the reduction of Cu^II^ to Cu^I^ in the presence of proteins under alkaline conditions, which results in the formation of bicinchoninic acid and a consequent intense purple color. The intensity measured is then utilized for the calculation of protein concentration in the solution by comparison with a calibration standard curve. In this particular assay, reduction of copper takes place due to the presence of amino acid residues present in proteins, which includes tyrosine, cysteine and tryptophan [112]. Contrary to other methods such as Lowry protein or Bradford assays, the universal protein backbone also contributes to the intensity of the color, minimizing the variability caused by differences on protein composition [113,114].

Regarding gel electrophoresis, this analytical technique has been widely employed for the separation and identification of proteins by their different molecular weight. During the separation process, proteins travel through a gel matrix, where smaller proteins run faster due to the low resistance they face in comparison with their larger counterparts [115,116]. However, this separation could be affected by the charge of the proteins. This problem is solved with the use of sodium dodecyl sulfate (SDS) and polyacrylamide gel, allowing a separation solely based on the polypeptide chain length [117]. SDS is a detergent that exhibits a great capacity for denaturing proteins by binding the protein backbone in a constant molar ratio. By using SDS together with a reducing agent, proteins are unfolded into linear chains with a negative charge proportional to the length of the polypeptide chain. This technique has been applied to the separation of proteins of the corona from the NM and classification by molecular weight. It is important to note that, although several approaches of digestion have been investigated, none of them are able to fully detach the complete amount of proteins from the NM surface because some proteins have interacted in an irreversible way [118]. This means that, even if this information is very valuable, there are some parts of the corona that are yet to be revealed.

Finally, qualitative and quantitative analysis of the PC is performed by LC-MS/MS. This technique has become an essential tool in the field of proteomics because it allows the separation as well as detection of numerous and different proteins within complex mixtures such as physiological fluids with a very high sensitivity [119]. One advantage of this technique is the use of a very small volume of sample, quite useful in the case of a limited amount of sample. This method employs label-free quantification to determine the relative amount of protein in a biological sample; therefore, no isotope of indicator molecule is needed. In a typical measurement, gel electrophoresis is utilized to separate the proteins from the NM. The bands containing the proteins are then excised and later digested prior to their introduction in the instrument. LC is carried out, allowing the separation of proteins according to their different molecular weight. When the sample reaches the spectrophotometer, it is ionized, and the resulting ions are sorted depending on their mass-to-charge ratio. In addition, it is important to adequately treat the data obtained from the LC-MS/MS, and for that purpose, specialized software programs are available. The most widely used software for data treatment of PC analysis has been Scaffold [120] and Progenesis [121]. The main difference between both relies on their quantification methods. Scaffold uses spectral counting, which is based on counting the number of spectra identified for a particular peptide in the sample, then integrating the result for all the measured peptides of the proteins that are quantified. Differently, Progenesis is based on ionic intensity, which extracts the intensity signal coming from the peptide allowing uncoupling between identification and quantification. It is important to note that ionic intensity is only possible due to the high-resolution power of Orbitrap analyzers (and some others of similar performance). Both approaches are very similar and, generally, no significant variability is expected.

New approaches are being developed, which are interesting the so-called intact protein expression spectrometry [122]. This technique aims for the analysis of intact proteins by LC-MS/MS for the detection and quantification of the full protein profile of the corona by means of a data reduction software.

### 5.2. In Situ Analysis of the Protein Corona

As mentioned before, although ex situ analysis of the PC reveals important information about this phenomenon, it fails when studying the process in real time, and, in general, the characteristics of the SC. In order to obtain a deeper understanding of the SC and of the formation of the PC, several authors have performed in situ measurements that allow them to obtain real-time information of protein–NM and protein–protein interactions in the chosen media, without the need of the purification step. In the following paragraphs we will present some of the techniques used for these in situ measurements and, in addition, we will include some isolation approaches for the SC. Although separation strategies are not properly in situ, they are included in this section because their main focus is the study of the SC.

#### 5.2.1. Techniques—In Situ Measurements.

DLS has been mentioned before as a technique employed for monitoring the changes on the hydrodynamic size of NMs before and after protein interaction. Although initially this technique could be used to study the interactions of proteins with NMs, the main limitation is posed by the presence of unbound proteins, which causes interferences with the scattering signal. This means that the sample must be purified; therefore, the SC would be lost and not detected. An alternative known as depolarized dynamic light scattering (DDLS) has been proposed in order to study biomolecule–NM interactions in the presence of complex media as the unwanted signals from unbound free proteins are suppressed [123]. This option was investigated and discussed in detail by Balog and co-workers [104] when investigating the biomolecular corona of gold NPs. This technique relies on the optical anisotropy of NMs, because many NMs do not display a perfect spherical shape and also exhibit an inhomogeneous crystalline structure. After illumination with a laser source, the biological matrix gives a strong signal due to the fluctuations and the presence of many additional biomolecules in the media. In contrast, the depolarized scattering of a biological matrix is virtually invisible, making the measurement of the NM size under the presence of a physiological and complex fluid viable.

Fluorescence correlation spectroscopy (FCS) is another technique that has been suggested for the study of protein–MN interactions and SC in real time, because there is no need for purification of the NM–PC complex [105,124]. FCS consists of a correlation statistical analysis of temporal fluctuations of fluorescence intensity. It provides information about the photo-physics behind the observed modifications in fluorescence intensity along with the diffusion behavior and concentration of fluorescent particles. The main advantage of FCS is it being based on fluorescent labelling, where the signal is exclusively emitted from fluorescent NMs, and hence in situ measurements are possible even in the presence of unbound proteins and other biomolecules in the media. This technique is employed to monitor the interaction, study the kinetic parameters of the process, as well as to monitor conformational changes of proteins upon interactions with NMs. Röcker et al. [125] explored this possibility on their investigation of polymer-coated FePt and CdSe/ZnS NPs incubated in a solution of human serum albumin (HSA). In their study, the authors reveal the formation of an HC of 3.3 nm thickness and the kinetic coefficients for HSA association–dissociation processes.

An interesting non-optical methodology for the study of protein–NM interaction in complex media was presented by Carril and co-workers [103]. In their publication, they proposed the alternative of monitoring the hydrodynamic radius of the NP by following changes of ^19^F diffusion by nuclear magnetic resonance (NMR), which was achieved by incubating NPs labelled with ^19^F in complex media. Diffusion NMR spectroscopic measurements can help to resolve different compounds spectroscopically in a complex mixture, hence providing insight about slight modifications on size or aggregation in relation to their diffusion coefficients. The authors were able to quantify the adsorption of proteins under a turbid environment, presenting this technique as an interesting approach for the measurement of size under in vivo conditions, even in the presence of cells.

Quartz crystal microbalance (QCM) is another technique that has been applied to the study of protein adsorption on NM surfaces. Additionally, known as quartz crystal nano-balance, this technique measures the mass variations through modifications of the frequency on a piezo-electric crystal where there are very small mass additions [126]. It can be used under vacuum and in the presence of a liquid. In particular, Di Silvio et al. [127] investigated the interaction of NPs with model lipid membranes under flow conditions. The exploration of HC and SC was performed by QCM with dissipation monitoring combined with neutron reflectometry. The results reported a permanent alteration on the lipid bilayer caused by the SC, contrary to the effect of free serum. In a similar study, Wang and co-workers [128] investigated the influence of the SC and the solution chemistry on interactions between silver NPs and lipid membranes by QCM with dissipation monitoring. In this case, results indicated that PCs, electrolyte concentration, and cation valence play an important role on silver NP stability and its consequent deposition on lipid membranes. Regarding the effect of the SC, it caused a reduction in the attachment of silver NPs on the lipid membranes.

Numerous additional techniques are being utilized for the study of the elusive SC and its influence on other processes. For example, Weiss et al. [17] presented an in situ study of the PC formed on silica microparticles based on the use of confocal laser scanning microscopy in combination with microfluidics. The study was focused on the time-evolution of the PC, and their approach enabled observation of an evolution from the HC to an additional SC. In addition, they reported the antifouling properties of a zwitterionic-functionalized NP, where only an SC was formed. Another in situ analysis was performed by Sanchez-Guzman et al. [129], where they aimed to reveal the molecular basis of SC formation. In their investigation, the authors investigated the PC formed by hemoglobin on SiNPs. They were able to study SC interactions by using synchrotron radiation circular dichroism and cryoTEM, showing that NPs alter the stability and structure of weakly bound proteins.

Although SC and weak protein interactions are difficult to observe and investigate, it has been demonstrated that their study is possible by using a wide variety of techniques and analytical tools. In situ approaches allow the study of weak interactions as well as real-time measurements, achieving information that can be complemented to ex situ studies for a more complete idea of the PC formation process and its consequences.

#### 5.2.2. Isolation Approaches

Although in situ strategies enable the obtaining of complementary information to ex situ approaches, the isolation of the SC still remains quite a complicated task due to the weak nature of these interactions. Nevertheless, some researchers have demonstrated that, although difficult, this separation can be performed by using an adequate strategy.

This is the case of Weber and co-workers [85], whose study focused on the preservation of the SC by using asymmetric flow-field fractionation (AF4) as a separation technique. This alternative allowed them to isolate SC proteins and to study their influence on cell uptake behavior. AF4 is a fractionation method that separates proteins, polymers, or NMs by its different size. When using this technique, a sample is loaded into the system and carried out under laminar flow to a separation chamber. Then, a separation field is perpendicularly applied against the sample flow. This will cause NMs or molecules present in the sample to be pushed towards the bottom of the channel. The molecules or NMs will diffused back at different extents, as a function of their Brownian motion, which is dictated by their characteristic size and would allow for a separation of NMs according to their sizes. In this particular case, increase in size of the NP is interpreted as the presence of a corona around the NP. Results indicated that the cell internalization process was not significantly affected by either of the coronas.

Finally, DCS has also been successfully employed for the detection and isolation of weakly bound proteins of the corona. In a publication by Davidson et al. [108], DCS was applied for the separation of both HC and SC of gold NPs upon the adsorption of bovine serum albumin (BSA). As mentioned before, DCS is able to separate NPs of similar density by mass, being able to differentiate between bare NPs and corona-coated NPs. In this study, the authors demonstrated the ability of DCS for the detection of very subtle changes and its potential use for SC studies.

### 5.3. Advantages and Disadvantages

The ex situ approach has been by far the most widely employed for the investigation of the PC phenomenon. This methodology allows for the isolation of the PC–NM complex from the free unbound proteins in the media, which is then characterized and analyzed. In this manner, proteins that display a high affinity for the surface and interact in an irreversible way with the material can be studied by LC-MS/MS. In general, this approach is relatively simple once you the adequate separation method is found, and it allows for a high control of the conditions of the experiment.

Nevertheless, this methodology usually fails at revealing information about the SC. Of lower stability and based on weaker protein–protein interactions, the SC is removed by use of the mentioned purification methods. It is important to consider that some authors have reported SC isolation by some techniques such as DCS [108] or AF4 [85]. In addition, the thermodynamic process of adsorption of some proteins has been carried out, but they are generally performed in simple solutions of single or few proteins. In addition, although it gives very useful information about the composition and characteristics of the corona, it does not obtain real-time data of interactions leading to PC formation.

In situ strategy, on the other side, it is known for giving information about the PC formation process in real-time and under more realistic conditions, where media and other biomolecules are present. This allows the investigation of protein–NM and protein–protein interactions and the observation of the evolution of the corona from a soft phase to a hard phase. SC can be studied by this methodology and researchers are working with several techniques for its isolation and further analysis. As has been mentioned before, techniques such as DDLS [104] or FCS are employed for the in situ investigation of the process, while DCS [108] and AF4 [85] have been reported as alternatives for SC isolation.

In summary, both approaches lead to interesting and relevant data regarding the process of PC formation and its composition. Although they have some drawbacks, their combination is a powerful methodology for the study of the PC. Research is still ongoing and additional techniques and strategies are being developed.

## 6. Protein Corona for Biomarker Discovery

A biomarker is generally described as a measurable indicator of modifications in a biological process and is currently used for disease detection. Discovery of biomarkers has typically been performed by analyzing blood samples of individuals with different illnesses by means of mass spectrometry-based proteomics. Nevertheless, this approach has proven to be very complex and challenging, and particularly affected by the high concentration of proteins present in the blood. This is due to two main reasons: (i) the low concentration of biomarkers in the biofluid, which needs to be over the signal-to-noise ratio limit of the instrument to be detected; and (ii) the effect of highly abundant proteins on the measurement.

PC of NMs has been proposed as a diagnostic tool with great potential for biomarker discovery due to the ability of nano-systems to interact with a great variety of proteins [51,130,131,132]. PC composition is influenced by the choice of biofluid; therefore, it is expected for differences to be observed between coronas formed on blood samples from healthy and ill patients. The first comparative study of PCs formed in the plasma of patients suffering from different diseases was performed by Hajipour et al. [98]. In their publication, the authors incubated polystyrene and silica NPs in human plasma of patients with different medical conditions: pregnancy, smoking, diabetes, common cold, fauvism, hemophilia, thalassemia, hypercholesterolemia, hypofibrinogenemia, rheumatism, and breast cancer. Their results indicated that the composition of PC was influenced by the human plasma samples of patients with different diseases, leading to the introduction of the term “personalized protein corona” as a determinant factor for nanomedicine. Similar studies have been carried out by using lipid NPs [99,100], metallic NPs [133], and graphene oxide nano-sheets [81], which support the existence of a personalized PC and prove that it can be potentially applied to biomarker discovery.

In addition, Hadjidemetriou and co-workers have proposed the use of the PC formed under in vivo conditions for biomarker discovery [50]. In their publication, the authors administered lipid-based NPs into mice bearing two different tumor models: subcutaneous melanoma (B16–F10) and human lung carcinoma xenograft (A549). Results demonstrated the advantages on the in vivo PC over its in vitro counterpart, as the corona formed after injection in mice displayed a wider range of proteins that were not detected solely by incubation in plasma. Their team also performed the first study of in vivo PC formed in humans in a proof-of-concept study [134]. The authors reported the analysis of the human in vivo PC formed on the surface of liposomes after administration of Caelyx (PEGylated doxorubicin-encapsulated liposomes), showing the potential of PC for biomarker discovery in humans.

These positive results motivated G. Caracciolo et al. [135] for the development of a platform for the detection and identification of diseases. In their work, the authors combined the concepts of disease-specific PC and sensor array technology for the creation of this platform. By the analysis of PCs of different NPs and from different plasma samples, Caracciolo and co-workers were able to create a unique fingerprint for each cancer type. This technology was tested by using plasma samples of patients undergoing treatment for different types of cancer (lung, brain, and pancreas), the outcome revealing the capacity of the platform to discriminate among different cancer types.

## 7. Conclusions

PC has become an important feature of NMs proposed for biomedical applications that must be understood prior to any translation into clinics. As a result of the interaction of NMs with biomolecules upon contact with a biological fluid, a corona of proteins is formed onto the surface of the NM, leading to a modification of its initial physico-chemical characteristics along with its behavior, fate, and performance. Protein adsorption that leads to this multi-layered protein structure is quite a complex process, where numerous parameters need to be taken into account such as incubation conditions, physiological fluid, NM properties, and many more.

Due to the complexity of the process, PC formation study is quite complicated. A wide variety of approaches and techniques have been employed for this purpose, leading to a deeper understanding of this phenomenon. Several studies have revealed the existence of what is known as HC and SC, which are formed depending on the affinity of proteins for the surface, the proteins present the solution, as well as the functionalization of the NM, among many others. Ex situ approaches are so far the most extensively utilized, and they allow isolation of the HC and the monitoring of changes in the physico-chemical properties of NMs after protein adsorption. On the other side, in situ methodologies reveal details about the PC formation process and allow a real-time study of protein–NP as well as weaker protein–protein interactions (SC). In the end, there is not one perfectly adequate methodology for PC analysis, although many combined experimental techniques lead to a more complete investigation.

During the last 15 years, researchers have been intensively working on unveiling the secrets behind PC formation and its consequences. Beginning with methodologies for its isolation and the analysis of its composition, researchers have taken a step forward towards its analysis under real-time conditions. Although it is still yet to be completely understood, the knowledge gained from multidisciplinary efforts has allowed scientists to take advantage of this process and start to exploit their potential for biomedical applications. New approaches and techniques are being developed and, surely, they will bring novel knowledge and scientific breakthroughs with them.

## Figures and Tables

**Figure 1 nanomaterials-11-00888-f001:**
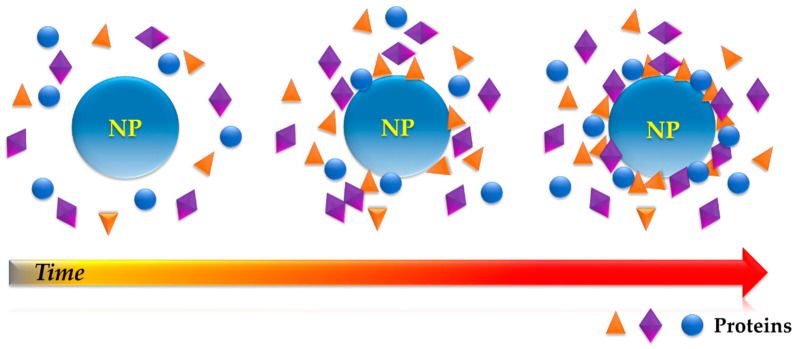
Protein adsorption onto a nanoparticle (NP) overtime leading to the formation of the so-called “protein corona” (cross-section view).

**Figure 2 nanomaterials-11-00888-f002:**
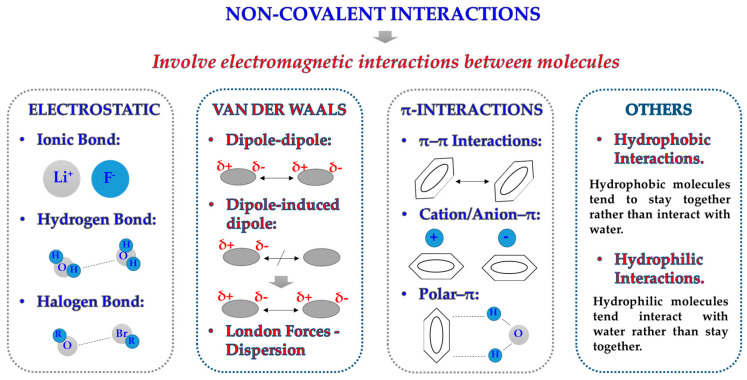
Schematic illustration of the different types of non-covalent interactions that could participate in the protein adsorption process.

**Figure 3 nanomaterials-11-00888-f003:**
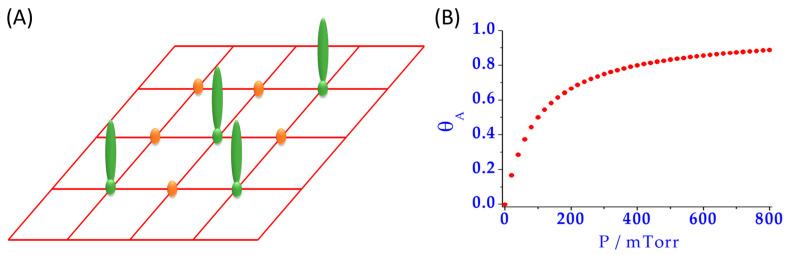
(**A**) Scheme of the equivalent free sites (orange) and occupied ones (green) assumed on the surface of an NM for the Langmuir model; (**B**) Example of a plot of the surface coverage (*θ_A_*) versus the partial pressure of the adsorbate (P), where the saturation point can be observed.

**Figure 4 nanomaterials-11-00888-f004:**
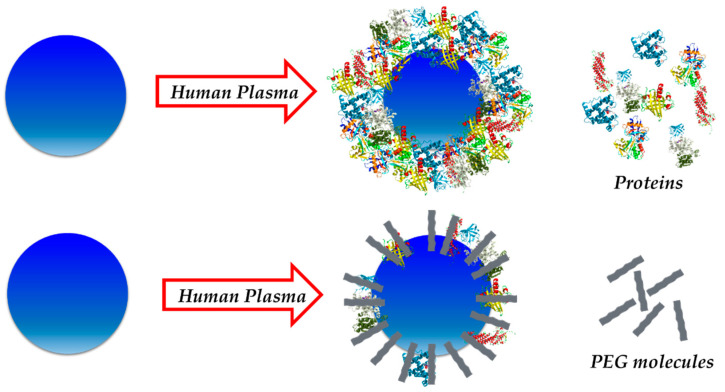
PEGylation of NPs minimizes interaction between proteins and NPs, leading to the formation of a smaller protein corona (PC) (cross-section view).

**Figure 5 nanomaterials-11-00888-f005:**
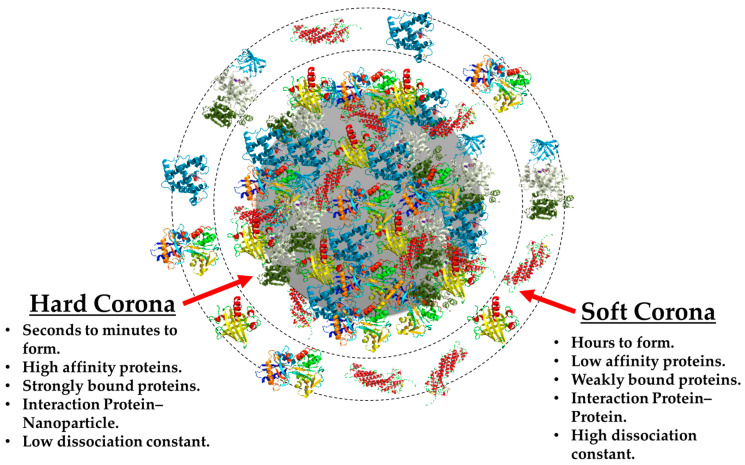
Schematic comparison of the main differences between the “hard” corona and the “soft” corona.

**Figure 6 nanomaterials-11-00888-f006:**
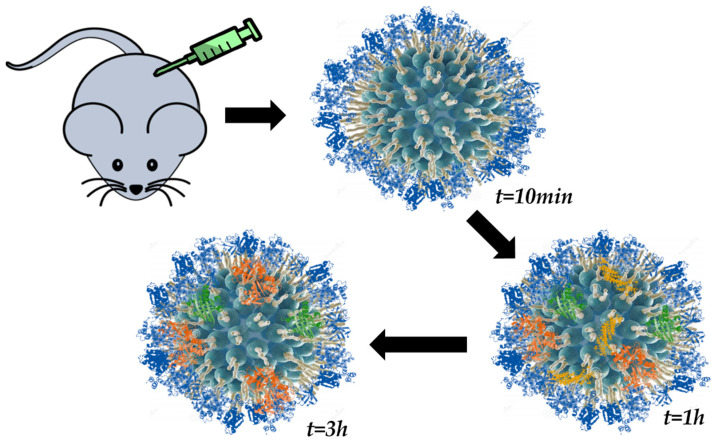
Schematic representation of the time-evolution study of the PC on lipid NPs performed by Hadjidemetriou et al. [74] (cross-section view).

**Figure 7 nanomaterials-11-00888-f007:**
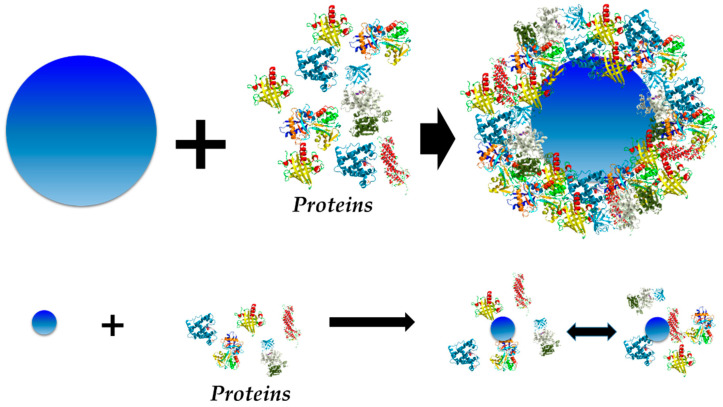
Schematic representation of the results obtained by Liu et al. [84] for 6–9 nm cerium oxide NPs—no HC formation, just SC—vs. the HC usually observed for larger NPs (cross-section).

**Figure 8 nanomaterials-11-00888-f008:**
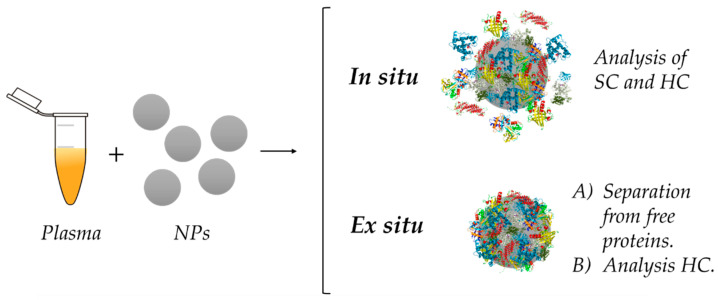
Schematic comparison of the in situ and ex situ approaches for PC investigations. NPs are incubated in plasma containing proteins, leading to formation of the corona. While ex situ analysis performs a purification step prior to analysis, in situ measurements are carried out in protein media without separation, which allows the study of both HC and SC.

**Figure 9 nanomaterials-11-00888-f009:**
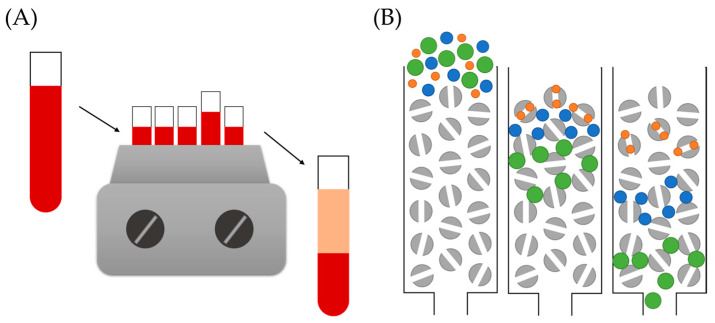
Two of examples of separation techniques employed for PC–NM complex isolation: (**A**) traditional centrifugation, where high-density materials form a pellet at the end of the tube while lighter components remain in solution; (**B**) size exclusion chromatography, where smaller substances follow a longer path than larger NMs, and hence are separated by size.

**Figure 10 nanomaterials-11-00888-f010:**
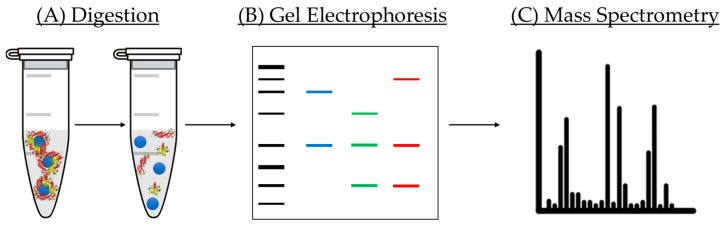
Scheme of the steps followed for analysis of the coronas of NMs: (**A**) digestion allows separation from proteins from the NM surface; (**B**) gel electrophoresis is used for separating proteins of different molecular weight; (**C**) ionized samples are separated by liquid chromatography and analyzed by mass spectrometry.

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
