# Peer review of "Hard and Soft Protein Corona of Nanomaterials: Analysis and Relevance"

_nanomaterials, 2021, doi:10.3390/nano11040888_

Round 1

Reviewer 1 Report

This review manuscript focuses on the topic of nanomaterial coronas. It is comprehensive, up to date and reasonably well written. I have, however, two major criticisms: 
1) The authors miss a trick by not describing the protein corona composition. What proteins are there, how do they vary as a function of nanomaterial properties (particularly composition and size) and what is the significance of protein diversity as a function of nanoparticle properties?
2) Section 4 is very disappointing and I would advise the authors to rewrite completely. The method description is highly inconsistent (some methods described in detail, others not), of variable quality and shows poor understanding of (at least some of) the methods. Section 4.1.3 is totally pointless. It's not about corona, but nanoparticle characterisation, and only describes a very small sub-set of relevant techniques. For example, what is the point of describing TEM? It cannot "see" the proteins. Why describe DLS but not NTA (aka PTA)?  Many other techniques are missing: UV-VIS, surface techniques (e.g.XPS), Synchrotron based techniques etc. Lines 549-559 read as part of a method section and don't match the style of the rest of the manuscript.
A minor point is that protein-nanomaterial interactions are described nicely in lines 75-80, but the schematic (Figure 2) bears no correspondence to the text. 
Also, shouldn't the title mention nanomaterials?
Stylewise, it is very strange that the figures are not referenced in the text and occasionally appear out of sequence.
Finally, some language attention required, a few (not comprehensive) examples are:  line 75 (susceptible to not of), line 99 (its global charge not charged), line 125 (look at, not to), line 134 (surfaces have, not has), line 135 (all being, not being all of them), line 192 (related to, not with), line 211 (adsorption to, not in)…….

Author Response

We would like to thank the Reviewer for their positive opinion of our manuscript and their feedback. In order to account for the limitations indicated by the reviewer, several changes in the manuscript have been made. Please see the attachment. 

Reviewer 2 Report

I would publish this interesting and informative piece after minor revision noted below:

1- Recent findings revealed that the combination of sensor array technology and protein corona (known as protein corona sensor array technology) may have a disease detection capacity. The readers might benefit from the role of sensory array protein corona for identification and discrimination of diseases.

 2- The purity of protein corona is a remarkable overlooked factor in the field; in fact, by ignoring the purity of protein corona, many publications count noise and protein aggregation as a real data in analyzing protein corona (e.g., Nat Commun 12, 573 (2021)). It might be interesting to add a section on the purity of protein corona and errors in proteomic analysis of the corona outcomes.

Author Response

We would like to thank the Reviewer for their positive opinion of our manuscript and their feedback. In order to account for the limitations indicated by the reviewer, some modifications have been made. Please see the attachment.

Reviewer 3 Report

The authors reviewed multiple relevant articles and summarized the findings on the formation of protein corona (both soft and hard) around nanomaterials (NMs) and their possible effect in cellular uptake, biodistribution and in biomedical applications in general. The authors recommended for need to characterize them correctly before using NMs for biomedical applications. I have some specific suggestions on this review article:  

  1. Figure 2: non-covalent interactions (II-interaction) is not clear? What this “II” refers to?
  2. Unnecessary reference citation where no references are needed. g Line 25:  In order to describe the protein-NM interaction, we must take a closer look to the different steps of the process itself [33].
  3. Line 160 ……..this process. It is not clear what “this process” refers to, is it surface functionalization or the PC for applications?
  4. Line 177”….it has been reported to cause no toxicity nor immune response while displaying high solubility and stability in water”. This sentence is not appropriate because anti-PEG antibody production in the context of NM has been reported. E.g. see this review. https://www.sciencedirect.com/science/article/pii/S0169409X20301083
  5. Line 331: “Employing this different type of procedure, scientists have been able to observed” please fix as  able to observe, NOT observed.

Author Response

(The authors gave the same response as above.)

Round 2

Reviewer 1 Report

The revised manuscript is substantially improved in style and content. I would still advise some careful reading, e.g. last sentence of section 6: 

This technology was tested by using plasma samples of patients undergoing different types of cancer (lung, brain and cancer), the outcome revealing the capacity of the platform to .......

One final comment: it appears that the authors use the words nanomaterial and nanoparticle interchangeably, and I would also advise to consider this terminology carefully in a final revision. 

Author Response

(The authors gave the same response as above.)
